# A Novel Endo-Polygalacturonase from *Penicillium rolfsii* with Prebiotics Production Potential: Cloning, Characterization and Application

**DOI:** 10.3390/foods11213469

**Published:** 2022-11-01

**Authors:** Meng-Jie Hao, Dan Wu, Yan Xu, Xiu-Mei Tao, Ning Li, Xiao-Wei Yu

**Affiliations:** 1Laboratory of Brewing Microbiology and Applied Enzymology, School of Biotechnology and Key Laboratory of Industrial Biotechnology of Ministry of Education, Jiangnan University, Wuxi 214122, China; 2State Key Laboratory of Food Science and Technology, Jiangnan University, 1800 Lihu Avenue, Wuxi 214122, China; 3Guangzhou Puratos Food Co., Ltd., Guangzhou 511400, China

**Keywords:** pectic oligosaccharides, endo-polygalacturonase, prebiotic, antibacterial, antioxidant

## Abstract

In this study, a potential producer of prebiotics, a novel endo-polygalacturonase pePGA from *Penicillium rolfsii* BM-6, was successfully expressed in *Komagataella phaffii*, characterized and applied to produce pectic oligosaccharides. The optimum temperature and pH of pePGA were 60 °C and 6.0. The purified recombinant enzyme showed a good pH stability and was stable from pH 3.5 to 8.0. The *K*_m_, *V*_max_ and *k*_cat_ values of pePGA were 0.1569 g/L, 12,273 μmol/min/mg and 7478.4 s^−1^, respectively. More importantly, pePGA-POS, the pePGA hydrolysis products from commercial pectin, had good prebiotic and antibacterial activities in vitro. The pePGA-POS was able to significantly promote the growth of probiotics; meanwhile, the growth of *Escherichia coli* JM109, *Staphylococcus aureus* and *Bacillus subtilis* 168 was effectively inhibited by pePGA-POS. In addition, pePGA-POS also had the DPPH radical scavenging capacity. These properties of pePGA-POS make pePGA attractive for the production of prebiotics.

## 1. Introduction

Pectic oligosaccharides (POS), considered as emerging potential prebiotic candidates [1], have multiple biological functions including anti-cancer activity [2], antibacterial activity [3,4] and antioxidant activity [5]. Previous studies showed that different sources of POS can promote the growth of probiotics [1,6,7]. Currently, POS are obtained by pectin degradation. Pectin is an important component of plant (all higher plants, gymnosperms, pteridophytes, bryophytes and Chara) cell walls [8]. It is mainly composed of homogalacturonan and rhamnogalacturonan-I [9]. The former constitutes approximately 60% of the pectin of plant walls. Homogalacturonan is formed by D-galacturonic acid residues linked by α-1,4 -glycosidic bonds. The D-galacturonic acid units are usually partially methyl esterified at C-6, and they are partially acetyl esterified at O-2 and/or O-3 in a few plant species [10].

The pectin-rich agricultural by-products are expected to be the sources of POS, such as olive pomace, sugar beet pulp, potato pulp, citrus waste and apple pomace [11]. POS can be produced by different methods, including physical degradation [12], chemical degradation [13], and enzymatic hydrolysis [14,15]. In particular, due to fewer byproducts, high selectivity and mild reaction conditions [3], enzymatic hydrolysis becomes a current research hotspot by using a series of pectinases to achieve the purpose of degrading pectin.

Pectinase is a class of specific enzymes that can degrade pectin substances. Pectinase has been widely used in the agricultural field, including tea and coffee processing, fruit processing, vegetable oil extraction and in the animal feed industry [16]. According to the cutting sites, pectinases can be divided into different classes, including pectate lyase (PL), pectinlyase (PNL), pectin esterase (PE) and polygalacturonase (PG) [17].

Polygalacturonase is one of the most widely studied pectinases comprising exo-polygalacturonase (exo-PGase, EC 3.2.1.67) and endo-polygalacturonase (endo-PGase, EC 3.2.1.15) [18]. Endo-PGases act randomly on unmethyl esterified polygalacturonic acid chains to produce oligomeric galacturonic acid, which is the main component of POS. Some endo-PGases with processive behavior were able to produce significant amounts of mono-galacturonic acid during enzymatic hydrolysis, which is not conducive to the accumulation of oligomeric galacturonic acid [19,20]. In recent years, several researchers reported that the endo-PGases from *Penicillium* sp. usually had high specific activities. Moreover, its hydrolysis products were mainly composed of oligomeric galacturonic acid with different degrees of polymerization and a minor amount of mono-galacturonic acid [21,22,23]. Therefore, the endo-PGases from *Penicillium* sp. seem to be appropriate to produce POS with high degrees of oligomeric galacturonic acid.

In this study, we discovered a novel endo-PGase derived from *Penicillium rolfsii* BM-6, named pePGA. PePGA was then expressed heterologously in *Komagataella phaffii* and characterized. Finally, the POS product enzymatically hydrolyzed by pePGA was analyzed, and it showed good prebiotic, antibacterial and antioxidant activities in vitro.

## 2. Materials and Methods

### 2.1. Strains, Vectors, and Chemicals

*P. rolfsii* BM-6, a highly polygalacturonases productive strain, was preserved in our laboratory. *Escherichia coli* Top 10 and *K. phaffii* X33 (formerly *Pichia pastoris X33*) were used as cloning host and expression host, respectively. The vector pPICZαA was used for gene cloning and protein expression. *E. coli* TOP10 Competent Cells were purchased from Sangon (Shanghai, China). *K. phaffi* X33 and pPICZαA were purchased from Invitrogen (Carlsbad, CA, USA). The pucTOPO-Blunt vector used in the DNA sequencing was purchased from Talen-bio (shanghai) Technology Co., Ltd. (Shanghai, China).

The restriction enzymes *Bgl* II, *EcoR* I and *Xba* I were purchased from TaKaRa (Dalian, China). The Plasmid Mini-PREPS Kit, PCR Product Purification Kit and 12.5% SDS-PAGE Color Preparation kit were purchased from Sangon (Shanghai, China). The substrate, polygalacturonic acid (P 3850), was purchased from Sigma (St. Louis, MI, USA). The pectin from apple (93854) and the pectin from citrus (P 9135) were purchased from Sigma (St. Louis, MI, USA). The commercial demethylated pectin (from citrus fruit) with 31% degree of esterification was purchased from Zhuoxin (Henan, China). The mannuronic acid oligosaccharides (MOS) and carrageenan oligosaccharides (COS) were purchased from Qingdao HEHAI Biotech Co., Ltd. (Shandong, China). D-(+)-Galacturonic acid monohydrate was purchased from Weng Jiang Reagent (Guangdong, China). Trigalacturonic acid and digalacturonic acid were purchased from Sigma (St. Louis, MI, USA). All other chemicals were chemically pure and purchased from Sinopharm Chemical Reagent.

### 2.2. Gene Cloning

*P. rolfsii* BM-6 was inoculated in the YPD liquid media at 30 °C and 200 rpm for 48 h to be used for the DNA and RNA extraction. The cDNA was obtained by RNA reverse transcription. According to the genome sequence of *P. rolfsii* (GenBank: QMFL01000022.1), the gene encoding a putative endo-PGase was amplified by polymerase chain reaction (PCR) using primers 5′-ATGCCTAAGCTCTTTAGTTCTCTTCTGCTAGCC-3′ and 5′-TCACGAGCAGCTAGCAACACTAGGCA-3′. The PCR products were inserted into the pucTOPO-Blunt vector for sequencing.

SignalP 5.0 (http://www.cbs.dtu.dk/services/SignalP (accessed on 26 October 2021) was used to predict the signal peptide. The gene encoding the mature enzyme was amplified by PCR using primers 5′-ACCGGAATTCTCACCCGTTGCCGAGCCAGC-3′ and 5′-GCTCTAGAGCTCTAGATTACGAGCAGCTAGCAACACTAGGCACG-3′. *EcoR* I and *Xba* I were used to digest the amplified fragment, and the digested DNA fragment was inserted into the vector pPICZαA between *EcoR* I and *Xba* I restriction sites.

### 2.3. Expression and Purification of the Recombinant Enzyme

The recombinant plasmid was linearized with *Bgl* II and transformed into *K. phaffii* X33 competent cells by electroporation. The positive clone was verified with primers α-factor (5′-ATACTACTATTGCCAGCATTGCTGC-3′) and 3′AOX (5′-TCAGGCAAATGGCATTCTGACATCC-3′). The preparation of the buffered glycerol-complex medium (BMGY) and buffered methanol-complex medium (BMMY), and the method for the induction of positive clone by methanol were according to the Multi-Copy Pichia Expression Kit (Invitrogen). The single clone was selected and inoculated into 25 mL BMGY medium. After overnight incubation at 30 °C and 200 rpm, the cells were collected by centrifugation at 4700× *g* for 5 min and transferred to 100 mL BMMY medium, cultured at 28 °C and 200 rpm for 120 h, and induced with methanol every 24 h.

After methanol induction, the cell debris was removed by centrifugation at 6300× *g* for 10 min at 4 °C. The crude enzyme was dialyzed in a 20 mM pH 7.0 PBS buffer for 24 h, and the enzyme was purified using a 5 mL PrePack Q Purose 6 HP exchange column (Qianchun Bio; Zhejiang, China). The purified recombinant pePGA was concentrated by a 10 kDa ultrafiltration centrifuge tube.

### 2.4. SDS-PAGE Analysis and Recombinant Protein Identification

The 12.5% SDS-PAGE Color Preparation kit was used to prepare sodium dodecyl sulfate-polyacrylamide gel electrophoresis (SDS-PAGE). The protein samples were mixed with the 4 × Protein SDS PAGE Loading Buffer (TaKaRa; Dalian, China) and heated in boiling water for 15 min for electrophoresis. The protein concentration was determined by a Bradford Protein Concentration Determination Kit. To verify the identity of the recombinant protein, the gel purified protein was digested by trypsin at 37 °C for 20 h and subjected to matrix assisted laser desorption ionization-time of flight-mass spectroscopy (MALDI-TOF-MS) analysis.

### 2.5. Enzyme Activity Assay

The endo-PGase activity was determined by using the 3,5-dinitrosalicylic acid (DNS) method [24]. All reactions contained 50 μL properly diluted enzyme solution and 450 μL of 0.33% (*w*/*v*) polygalacturonic acid dissolved in 50 mM pH 6.0 Na_2_HPO_4_-citric acid buffer at 60 °C. The reaction system was incubated for 10 min under the standard condition. Next, 750 μL DNS reagent was added to terminate the reaction. The mixture was boiled for 5 min and quickly cooled to room temperature with ice water. Finally, the absorbance at 540 nm was measured. All experiments were performed in triplicate.

One unit (U) of endo-PGase was defined as the amount of enzyme that released reducing sugars equivalent to 1 μmol of D-(+)-galacturonic acid per min under the standard condition.

### 2.6. Biochemical Characterization of the Purified Recombinant Enzyme

The optimum pH was measured from 3.0 to 8.0 at 20 mM Na_2_HPO_4_-citric acid buffer at 55 °C. To determine the pH stability, the enzyme was treated in 0.2 M Na_2_HPO_4_-citric acid buffer with different pH (3.5–8.0) at 25 °C for 1 h as pretreatment. The residual activity was determined under the standard condition, 60 °C at 50 mM pH 6.0 Na_2_HPO_4_-citric acid buffer for 10 min. To estimate the optimum temperature, the enzyme activity at temperatures ranging from 20 to 80 °C was measured. The thermal stability of the enzyme was determined by measuring the residual activity under the standard condition after pre-incubating the enzyme at 40, 45, and 50 °C in different periods (0–60 min).

Enzyme kinetic parameters were estimated with 0.4–2.5 mg/mL polygalacturonan acid as the substrate under 60 °C at 50 mM pH 6.0 Na_2_HPO_4_-citric acid buffer for 3 min. The enzyme kinetic parameters were calculated according to Michaelis–Menten equations. The software OriginPro 2022b was used to calculate the *K*_m_ and *V*_max_.

The specific activity of pePGA on the 0.33% polygalacturonic acid and two kinds of pectins (the pectin from apple with 55.3% degree of esterification and the pectin from citrus with 64.8% degree of esterification) were determined to test the substrate specificity.

All experiments were performed in triplicate.

### 2.7. Analysis of Hydrolysis Products

One percent polygalacturonic acid and pePGA (1 U/mL) were incubated at 40 °C for different periods. The hydrolyzed samples were filtered with 0.22 μm filter membrane, and then passed through a 3 kDa ultrafiltration concentration tube for purification. The filtered liquid was collected for analysis by the high-performance anion-exchange chromatography with pulsed amperometric detection (HPAEC-PAD), using a DIONEX ICS-5000+SP-5 (Waltham, MA, USA) instrument equipped with a CarboPac PA20 (3 mm × 150 mm; Waltham, MA, USA) maintained at 30 °C. The mobile phases included mobile phase A (deionized water), mobile phase B (1 M sodium acetate), and mobile phase C (250 mM sodium hydroxide).

Thin layer chromatography (TLC) was used to detect the components of the hydrolysis products by pePGA on the commercial demethylated pectin. One hundred milliliters of commercial demethylated pectin (4% *w*/*v*) and 4000 U pePGA were incubated at 40 °C for different times, and the samples were boiled for 10 min in water to inactivate the enzymes. The insolubles were eliminated by centrifugation at 8000× *g* for 5 min. Five microliters of properly diluted product solution was developed on a Merck silica gel 60 F_254_ plate twice with a mobile phase of n-butanol: acetic acid: water at a ratio of 2:1:1 (*v*/*v*/*v*). The solution containing 0.5% thymol (*w*/*v*), 5% concentrated sulfuric acid, 91% ethanol, and 4% water was used to detect the products. The silica gel plate was sprayed with the solution and heated to visualize the products.

The high-performance gel filtration chromatography (HPGFC; 1525 EF, Waters Corporation, Milford, CT, USA) was used to measure the average molecular masses of the polygaluronic acid, the commercial demethylated pectin, and the hydrolysis products, which was equipped with a differential refractive index detector and a column Ultrahydrogel TM (300 mm × 7.8 mm), which controlled the flow rate at 0.5 mL/min at 40 °C. We selected 0.1 N NaNO_3_ as the mobile phase and used dextran and glucose as the standards.

### 2.8. In Vitro Antimicrobial, Antioxidant and Prebiotic Activities of pePGA-POS

One hundred milliliters of commercial demethylated pectin (4% *w*/*v*) and 4000 U pePGA were incubated at 40 °C for 5 h. The mixture was boiled for 20 min in water to inactivate enzymes. Next, pePGA-POS was obtained by centrifuging the mixture at 6300× *g* for 20 min and freeze-drying the supernatant.

#### 2.8.1. Prebiotic Activity

Three kinds of microorganisms, *Pediococcus acidilactici, Lactobacillus plantarum* and *Lactobacillus paracasei* were stored at −80 °C in Man-Rogosa-Sharpe (MRS) medium. The MRS medium contained 5 g/L of yeast extract, 10 g/L of peptone, 20 g/L of glucose, 10 g/L of beef extract, 1 g/L of tween 80, 2 g/L of K_2_HPO_4_, 5 g/L of sodium acetate, 0.5 g/L MgSO_4_·7H_2_O, 0.2 g/L MnSO_4_·H_2_O, and 2 g/L of triammonium citrate. The microorganism was inoculated in MRS agar plates at 37 °C for 48 h to be activated. The single clone was then inoculated into 5 mL MRS liquid medium overnight at 37 °C. The overnight culture of each microorganism 1% (*v*/*v*) was added to the tubes containing MRS mediums with different carbon sources (10 g/L glucose, 10 g/L pePGA-POS, 10 g/L fructooligosaccharides (FOS) or no carbon source). The mixture was inoculated at 37 °C for 36 h and the absorbance at 600 nm was measured at 0 h, 6 h, 12 h, 24 h and 36 h, respectively. All experiments were performed in triplicate.

#### 2.8.2. Antimicrobial Activity

To determine the antibacterial activity of pePGA-POS, the diameters of inhibition zone using the agar punch method were measured. *E. coli* JM109, *Staphylococcus aureus*, *Bacillus subtilis* 168 and *Candida albicans* were selected as indicator strains. The proper volumes of overnight cultures of microorganisms were respectively mixed with lysogeny broth (LB) solid medium to a final concentration of approximately 10^5^–10^7^ CFU/mL and the mixture was poured onto plates. After coagulation, 0.6 cm holes in the bacteria culture plates were made using hole punches, and then each was added with 50 μL of 400 mg/mL pePGA-POS, 1 mg/mL ampicillin and deionized water, respectively. The inhibition zone diameters were measured after incubation at 37 °C for 16 h to 24 h.

As an important laboratory parameter, the minimum inhibitory concentration (MIC) was identified as the lowest concentration of pePGA-POS to inhibit the visible growth of microorganisms. MIC was determined by the method of broth dilution according to Li et al. [3] with some modifications. The different final concentrations (150, 100, 50, 25, 12.5, 6.3, 0 mg/mL) of pePGA-POS in LB medium were prepared. The microbial inoculum was then added to the pePGA-POS broth dilution up to approximately 10^6^ CFU/mL, and the mixture was incubated at 37 °C for 18 h. The absorbance at 600 nm of all treatments were measured to confirm MIC. The treatments without pePGA-POS were termed as control and all experiments were performed in triplicate.

#### 2.8.3. DPPH Radical Scavenging Activity

The activity of pePGA-POS to scavenge the DPPH radical was measured according to the method described by Yu et al. [25], with modifications. The pePGA-POS was dissolved in deionized water to various concentrations from 0 to 60 mg/mL. Two milliliters of pePGA-POS solution in different concentrations and two milliliters of DPPH reagent (0.2 mM in ethanol) were mixed and incubated at 25 °C for 30 min in the dark. The absorbances of samples at 517 nm were measured to characterize the DPPH radical scavenging ability. The ascorbic acid solution was selected as a positive control. The scavenging rate was calculated by the following formula.
(1)$$Scavenging\;rate\;\left(\%\right)=\frac{A_{0}-\left(A_{i}-A_{j}\right)}{A_{0}}\times 100\%$$
where $A_{0}$: absorbance of blank control (deionized water instead of pePGA-POS solution), $A_{i}$: absorbance of the mixture of DPPH reagent with pePGA-POS solution or ascorbic acid solution, $A_{j}$: absorbance of pePGA-POS solution control (ethanol instead of DPPH reagent).

## 3. Results and Discussion

### 3.1. Expression and Purification of pePGA

The cDNA fragment was 1143 bp, encoding a peptide of 380 amino acids. The peptide shared 100% identity with a putative endo-PGase from *P. rolfsii* (GenBank: KAF3392764.1). According to the predicted outcome from SignalP 5.0, the signal peptide (MPKLFSSLLLAALAVGVIA) was removed and the gene sequence encoding the mature peptide was cloned in pPICZαA fused to the α-factor secretion signal to direct the export of the protein to the BMMY medium. The recombinant enzyme was named pePGA and successfully expressed in *K. phaffii* X33. After 120 h of methanol induction, the polygalacturonase activity of culture supernatant was 1571.7 U/mL.

The crude enzyme was purified by an ion exchange column and ultrafiltration centrifugation to electrophoretic homogeneity (Figure 1). Compared with the theoretical calculated molecular mass (36.56 kDa), the actual protein molecular mass (approximately 44.3 kDa) was higher. According to this phenomenon, some researchers speculated that the increase in molecular mass was due to O-glycosylation, N-glycosylation and acetylation modifications in *K. phaffii* [26]. The bands on the protein gel were cut off and digested with trypsin. Four peptides (LSDLPDDT, GYKEWSGPLLQ, DMTINNEAGDSAGGHNTDGFDIG, YDGGDLEGTPTSGIP) of pePGA were identified by MALDI-TOF-MS analysis. The results confirmed that the recombinant enzyme was correctly expressed in *K. phaffii* X33.

### 3.2. Biochemical Characterization of the Recombinant pePGA

The activity on polygalacturonic acid of the enzyme without pre-treated was set as 100%. The optimal pH of pePGA is 6.0 (Figure 2A). The enzyme retained more than 70% of the maximal enzyme activity at the range of pH 5.0–6.5 and almost completely lost activity at pH 7.5. The enzyme was very stable between pH 3.5 and 8.0 at 25 °C for 1 h with residual activity of almost 90–100% (Figure 2B). The optimal temperature of pePGA was 60 °C (Figure 2C). The enzyme was thermostable at 40 °C for 1 h and retained approximately 70% activity after incubating at 45 °C for 1 h. However, the enzyme activity dropped rapidly at 50 °C and the residual enzyme activity after treatment for 20 min was about 30% of the initial enzyme activity (Figure 2D). The *K*_m_, *V*_max_ and *k*_cat_ values of pePGA (Appendix A) were 0.1569 g/L, 12,273 μmol/min/mg and 7478.4 s^−1^.

The optimum pH and temperature for endo-PGases vary substantially. Most endo-PGases are slightly acidic enzymes with an optimum pH of 2.5–6.0 and an optimum temperature of 30–70 °C [17,23]. Endo-PGases had demonstrated good stability to pH. For example, endo-PG I from *Achaetomium* sp. Xz8 retained more than 75% activity after incubating at pH 3.5–8.0 for 1 h [27]. AnEPG from *Aspergillus nidulans* retained more than 62.7% activity at pH 2.0–12.0 after 120 h [28]. Endo-PG I from *Penicillium* sp. CGMCC 1669 retained over 60% activity after incubating at pH 2.0–6.0 for 1 h [29]. In this study, pePGA showed much better pH stability. Regarding the temperature stability, AnEPG retained 80.7% activity at 40 °C for 120 min and retained only 39.1% activity at 50 °C for 15 min [28]. PoxaEnPG28B-Pp from *Penicillium oxalicum* CZ1028 retained 100% activity at 50 °C and 60% activity at 55 °C for 1 h [23]. However, pePGA was only stable below 45 °C, which needs to be further improved by molecular engineering.

The pePGA showed the highest specific activity on polygalacturonic acid (8881.5 U/mg, 100%), followed by the pectin from apple (739.0 U/mg, 8.16%) and the pectin from citrus (583.1 U/mg, 6.56%). Several studies reported that the endo-PGases showed the highest activity towards polygalacturonic acid compared to the pectins, with a degree of methylation from 34–85% [27,29], which was consistent with the finding of our study.

### 3.3. Analysis of Hydrolysis Products

Polygalacturonase is a class of glycoside hydrolases (GH28) including exo-PGase and endo-PGase. Exo-PGase hydrolyzes polygalacturonic acid and releases mono-galacturonic acid [30], however, endo-PGase acts randomly on the galacturonic acid chains producing oligomeric galacturonic acid with different degrees of polymerization. The average molecular mass of polygalacturonic acid was approximately 411.0 kDa, and the polygalacturonic acid was hydrolyzed to substances with average molecular masses lower than 2.9 kDa after pePGA (1U/mL) treatment for 2 h (Appendix A). The hydrolysis products of pePGA from polygalacturonic acid were measured by HPAEC-PAD. As shown in Figure 3, the content of tri-galacturonic acid increased continuously over time, while di-galacturonic acid only accumulated in a small amount and the obvious accumulation of mono-galacturonic acid were not observed. Due to lack of patterns, the oligomeric galacturonic acid higher than three were detected but not identified (Appendix A).

The average molecular mass of the commercial demethylated pectin was approximately 728.1 kDa, and the commercial demethylated pectin was hydrolyzed to substances with average molecular masses lower than 2.4 kDa after pePGA (4000 U/100 mL) treatment for 5 h (Appendix A). As shown in Figure 4, at the beginning of the hydrolysis (2–30 min), the commercial demethylated pectin was hydrolyzed into oligomeric galacturonic acid and minor amounts of mono-galacturonic acid. This indicated that pePGA was a non-processive enzyme [19,20]. When the reaction was prolonged to 5 h, the commercial demethylated pectin was mainly hydrolyzed into di-galacturonic acid and tri-galacturonic acid, and a small amount of mono-galacturonic acid. These results by HPAEC and TLC indicated that pePGA was a typical endo-PGase and mainly released oligosaccharides with polymerization degrees higher than two depending on the source of the substrates.

The POS products hydrolyzed by various sources of endo-PGases have different degrees of polymerization. The endo-PGase PoxaEnPG28A from *Penicillium oxalicum* CZ1028 mainly hydrolyzed polygalacturonic acid into oligomeric galacturonic acid with polymerization degrees higher than three [22]. The major products of an endo-PGase PG1 from *Penicillium occitanis* were tri-galacturonic acid and tetra-galacturonic acid [21]. The hydrolysis products on polygalacturonic acid by an endo-PGase NfPG5 from *Neosartorya fischeri* P1 were mainly composed of mono-galacturonic acid, di-galacturonic acid and tri-galacturonic acid [31]. A previous study [32] reported that the major products of an endo-PGase from *Pectobacterium carotovorum* were tri- and hexa- galacturonic acid, and the endo-PGase was speculated to require substrates with degrees of polymerization higher than six to form a productive binding for hydrolysis.

### 3.4. Prebiotic Activity of Hydrolysis Products

Currently, FOS has been commonly used as prebiotics in many fields [33], which was used as a positive control in our study. As shown in Figure 5, the growth of probiotics was significantly promoted by pePGA-POS compared with the negative control (H_2_O, without carbon source). Notably, pePGA-POS was more conducive to the growth of *P.*
*acidilactici* than FOS, and was comparable to glucose. The POS produced by enzymatic hydrolysis from orange peel wastes was able to augment the growth of *Bifidobacterium.*
*infantis* (ATCC 15697) and *L.*
*acidophilus* (AS 1.1854); however, FOS showed higher efficiency than POS [3], which is in line with the results of *L.*
*plantarum* and *L.*
*paracasei* in our study.

In previous studies, it was confirmed that the POS from different sources could promote the growth of probiotics to varying degrees. The POS with degrees of polymerization between 3 to 7 from bergamot by enzymatic hydrolysis could promote the growth of *Bifidobacteria* and *Lactobacilli*, meanwhile inhibiting the growth of *Clostridia* [6]. The POS from citrus peel pectin by chemical degradation could be metabolized by *L. Paracasei* LPC-37 and *B. Bifidum* ATCC 29521 [13]. The POS from apple pectin by physical degradation had the ability to increase the number of *Bifidobacteria* and *Lactobacilli*, in the meanwhile, to decrease the number of *Clostridia* and *Eubacteria* [12].

With the continuous expansion of the market demand of prebiotics, the POS from relatively cheaper sources such as pectin-rich agricultural by-products would be attractive for the prebiotic market [11]. Due to the prebiotic activity of pePGA-POS, pePGA was confirmed to have the ability to provide an alternative source of prebiotics from pectin.

### 3.5. Antimicrobial Activity of Hydrolysis Products

The diameters of inhibition zones were selected to reveal the antimicrobial activity of pePGA hydrolysis products against representative bacteria, including gram-negative bacteria (*E.*
*coli* JM109) and gram-positive bacteria (*S.*
*aureus, B.*
*subtilis* 168). The results are illustrated in Figure 6. The enzymatic hydrolysis products showed obvious antibacterial activity against bacteria. The inhibition zone diameters of *S. aureus*, *E. coli* JM109, and *B.*
*subtilis* 168 were 21.2 mm, 11.3 mm and 10.1 mm, respectively, while the hydrolysis products had almost no inhibition against *C.*
*albicans.* Ampicillin is one of the most important antibiotics belonging to the β-lactam group, which acts on bacteria and certain parasites [34]. Therefore, it was consistent with our results that ampicillin did not have an inhibitory effect on *Candida albicans* (Figure 6), which belongs to the fungus. Li et al. [3] reported that POS from orange peel wastes showed pronounced inhibitory activity against bacteria, but showed no effect on fungi, which was consistent with the findings of our study.

In Table 1, the MIC value against *E.*
*coli* JM109 and *B.*
*subtilis* 168 were both 50 mg/mL, and the MIC value against *S.*
*aureus* was 12.5 mg/mL. Some researchers [3,4,35] showed that gram-positive bacteria were more sensitive to POS than gram-negative bacteria. However, in this study, the growth of *E.*
*coli* JM109 seemed to be more susceptible to inhibition by pePGA-POS compared with *B.*
*subtilis* 168. We speculate that although *B.*
*subtilis* 168 belongs to gram-positive bacteria, its sensitivity to pePGA-POS varies among species.

The mechanism of antibacterial activity of POS has not been well elucidated at present. By testing the antibacterial activity of POS and mono-galacturonic acid at different pH values (Appendix A), we speculated that the pH value was an important factor affecting their antibacterial activities. PePGA-POS and mono-galacturonic acid showed excellent antibacterial activities at pH 3.0–3.5 and pH 2.0, respectively. However, PePGA-POS and mono-galacturonic acid lost the antibacterial activities at pH 5.0 and pH 3.0–3.5, respectively. It was affirmed that the oligomeric galacturonic acid in pePGA-POS, instead of mono-galacturonic acid, played an antibacterial role. The oligosaccharides from other sources, such as mannuronic acid oligosaccharides and carrageenan oligosaccharides, [36,37] were reported to possess excellent antibacterial activities in some studies. However, the MOS and COS had no antimicrobial activity in this study (Appendix A). Xue et al. [4] reported that the POS with a degree of polymerization of 2–4 from mango peel wastes by enzymatic hydrolysis demonstrated the highest antimicrobial activity. Wu et al. [38] believed that antibacterial activity was related to membrane permeability, molecular mass and degree of methylesterification of POS. Wang et al. [35] suggested that antimicrobial activity was associated with the side chain methyl ester modification of POS. However, the antibacterial mechanism of methylated POS has not been elucidated. Further research on the antibacterial mechanism of POS could contribute to seeking new antibacterial substances and improving the added value of pectin-rich agricultural by-products.

### 3.6. DPPH Radical Scavenging Activity of Hydrolysis Products

DPPH is a stable free radical and has been widely used to evaluate the antioxidant activity of compounds. When DPPH is incubated with antioxidants, the absorbance at 517 nm would be decreased because the unpaired valence electron of the nitrogen atom in DPPH was reduced [39,40]. As shown in Figure 7, although the DPPH radical scavenging activity of pePGA-POS was weaker than ascorbic acid, the scavenging rate increased from 32.97% to 73.97%, with the concentration of pePGA-POS increasing from 10 mg/mL to 60 mg/mL, indicating the concentration-dependent effect of pePGA-POS on DPPH radical scavenging.

Previous studies [41,42,43] suggested that polysaccharides from various sources had good DPPH scavenging activity. Yu et al. [25] demonstrated that FCPOS-1 (a polysaccharide enzymatic extracted from finger citron pomace) at the concentration of 10 mg/mL had the highest scavenging rate of 94.07%, which was better than the scavenging activity of polysaccharide obtained from finger citron using alkali extraction. The more activated hydroxyl groups, higher galacturonic acid contents, smaller molecular mass and better water solubility of FCPOS-1 have been speculated to be the causes of the high scavenging activity. Other researchers believed that the DPPH radical scavenging activity was related to the free galacturonic acid contents and specific structures of POS [44].

In this study, the commercial demethylated pectin was hydrolyzed by pePGA to produce pePGA-POS. The final pePGA-POS product exhibited properties of lower molecular mass, exposure of more activated hydroxyl groups, and enhanced water solubility, resulting in the good ability of DPPH radical scavenging.

## 4. Conclusions

A novel endo-PGase named pePGA from *P.*
*rolfsii* BM-6 was cloned and successfully expressed in *K.*
*phaffii.* The purified pePGA was characterized, and its excellent stability over a wide range of pH allows the applications of pePGA in many aspects. More importantly, the hydrolysis products of pePGA contained mainly di-galacturonic acid and tri-galacturonic acid, indicating that pePGA was suitable for the production of POS with high degrees of oligomeric galacturonic acid. PePGA-POS obtained by pePGA hydrolysis of commercial demethylated pectin showed good antibacterial, antioxidant and prebiotic activities in vitro. PePGA-POS was able to effectively inhibit the growth of bacteria, particularly *S.*
*aureus.* The DPPH radical scavenging rate reached 73.97% when the concentration of pePGA-POS was 60 mg/mL. The growth of probiotics was significantly promoted by pePGA-POS; meanwhile, pePGA-POS was better than FOS in promoting the growth of *P.*
*acidilactici.* In this study, pePGA was confirmed to have the ability to provide an alternative source of prebiotics from pectin due to the favorable characteristics of pePGA-POS.

## Figures and Tables

**Figure 1 foods-11-03469-f001:**
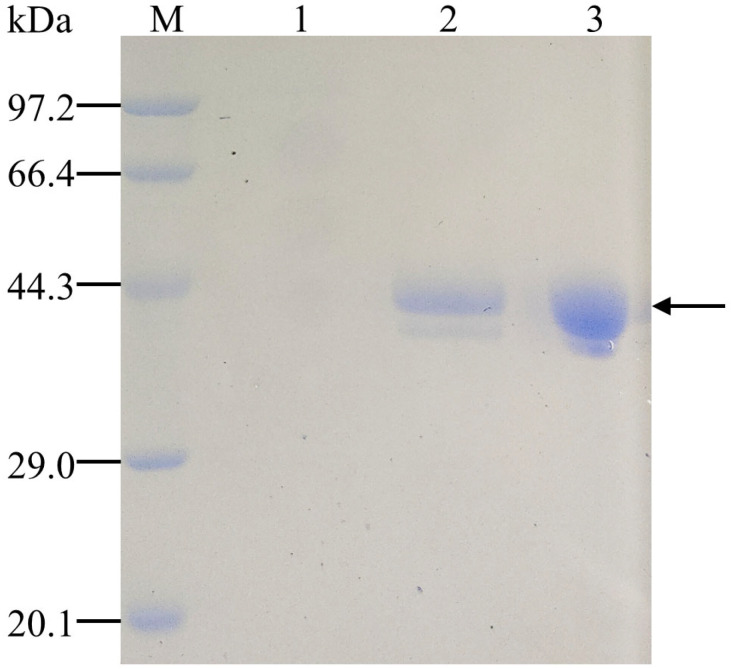
SDS-PAGE analysis of pePGA. Lane M, the standard protein markers (low); lane 1, the culture supernatant of empty vectors; lane 2, the culture supernatant of pePGA; lane 3, purified recombinant pePGA.

**Figure 2 foods-11-03469-f002:**
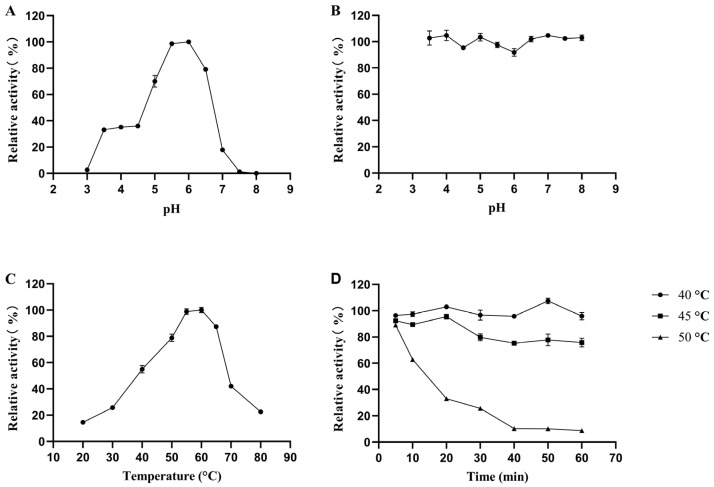
Biochemical characterization of the purified recombinant pePGA. (**A**) The effect of pH on enzyme activity. The enzyme activity was measured at 55 °C in 20 mM Na_2_HPO_4_-citric acid buffer from 3.0 to 8.0. (**B**) pH stability of pePGA. The enzyme was incubated in 0.2 M Na_2_HPO_4_-citric acid buffer with different pH (3.5–8.0) at 25 °C for 1 h and the residual activities were determined under the standard condition. (**C**) The effect of temperature on enzyme activity. The enzyme activity was measured at temperatures ranging from 20 to 80 °C in a 50 mM pH 6.0 Na_2_HPO_4_-citric acid buffer. (**D**) Temperature stability of pePGA. The enzyme was incubated at 40 °C, 45 °C and 50 °C in a 50 mM pH 6.0 Na_2_HPO_4_-citric acid buffer for various periods, and the residual activities were determined under the standard condition. Error bars present the standard deviation (*n* = 3).

**Figure 3 foods-11-03469-f003:**
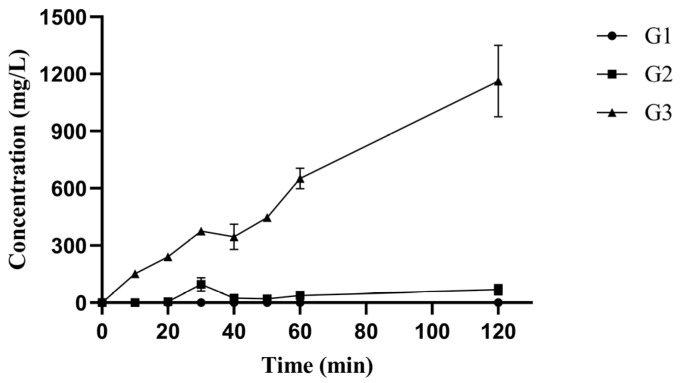
HPAEC-PAD analysis of products by pePGA on polygalacturonic acid. G1: mono-galacturonic acid, G2: di-galacturonic acid, G3: tri-galacturonic acid.

**Figure 4 foods-11-03469-f004:**
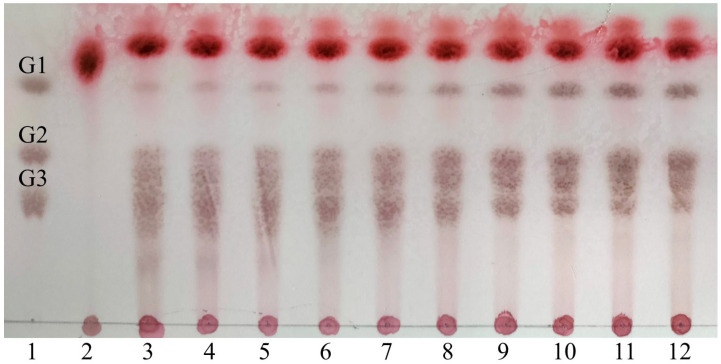
TLC analysis of the hydrolysis products of pePGA on commercial demethylated pectin. Lane 1, mixture of mono-galacturonic acid (G1), di-galacturonic acid (G2) and tri-galacturonic acid (G3); lane 2–12, the products at 0 min (lane 2), 2 min (lane 3), 5 min (lane 4), 10 min (lane 5), 20 min (lane 6), 30 min (lane 7), 1 h (lane 8), 2 h (lane 9), 3 h (lane 10), 4 h (lane 11) and 5 h (lane 12).

**Figure 5 foods-11-03469-f005:**
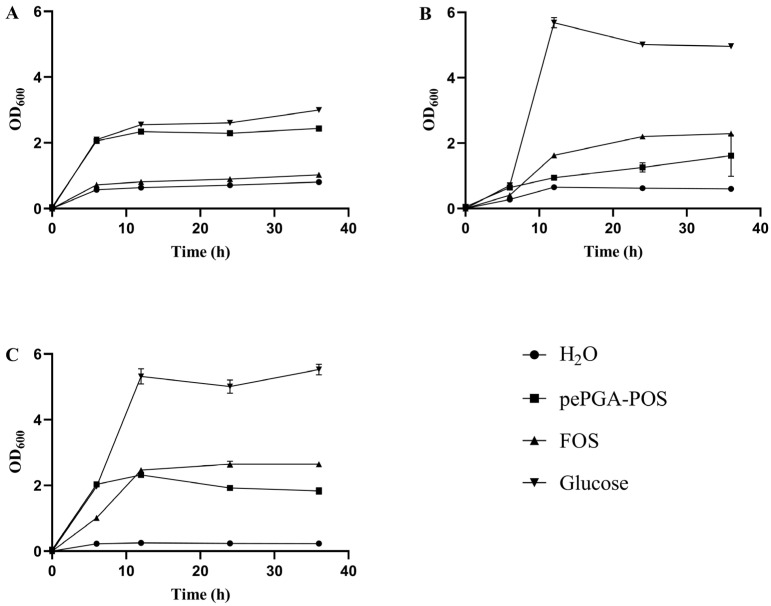
Growth curve of (**A**) *P. acidilactici*, (**B**) *L. paracasei*, (**C**) *L. plantarum* in different carbon sources or no carbon source.

**Figure 6 foods-11-03469-f006:**
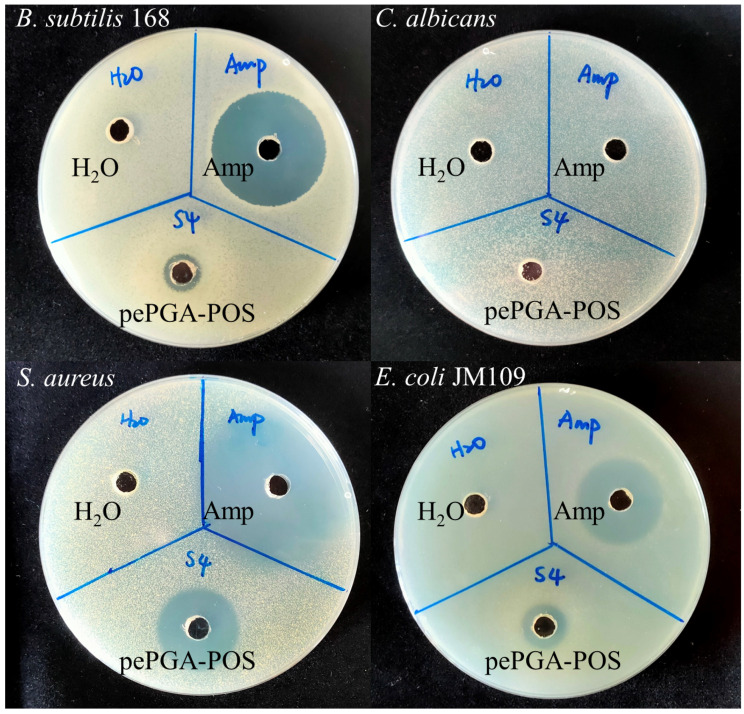
Antibacterial activities of pePGA-POS against *B.*
*subtilis* 168, *C.*
*albicans*, *S.*
*aureus* and *E.*
*coli* JM109.

**Figure 7 foods-11-03469-f007:**
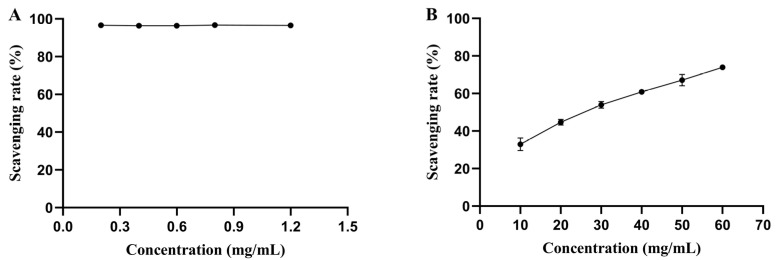
(**A**) The DPPH radical scavenging activity of ascorbic acid. (**B**) The DPPH radical scavenging activity of pePGA-POS.

**Table 1 foods-11-03469-t001:** The minimum inhibitory concentration (MIC) of pePGA-POS to inhibit the growth of *E. coli* JM109, *B. subtilis* 168 and *S. aureus.*

Antimicrobial Activities/MIC (mg/mL)	pePGA-POS
*E. coli* JM109	50
*B. subtilis* 168	50
*S. aureus*	12.5

## Data Availability

Data are contained within the article.

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
