# Peer review of "A Novel Endo-Polygalacturonase from Penicillium rolfsii with Prebiotics Production Potential: Cloning, Characterization and Application"

_foods, 2022, doi:10.3390/foods11213469_

Round 1
Reviewer 1 Report
The work entitle: “A novel endo-polygalacturonase from Penicillium rolfsii with prebiotics production potential: cloning, characterization and application” shows the characterization of a new fungal enzyme of potential biotechnological interest. It is well structured and presents interesting results that deserve to be published. However, there are several aspects that must be included, expanded and/or modified to facilitate the understanding of the data presented.
11.Despite the fact that the yeast Pichia pastoris has been reassigned to the genus Komagataella phafii after phylogenetic analysis of gene sequences, it still seems useful to indicate, at least in parentheses and in the materials and methods section, that it was Pichia pastoris since there are a greater number of references that denominates this organism with this previous designation, e.g. to include on Ln 68 behind K. phaffi X33 (formerly Pichia pastoris X33). Besides to include where this strain comes from/its supplier, what protocol/manual is used for its handling (Ln 95)/induction by methanol, etc. Perhaps it came originally from Invitrogen, since media BMGY (Ln 97) or BMMY (Ln 99) are mentioned without including their composition. I also miss the supplier of the E. coli strain used in the general DNA manipulations (Ln 68), that for the vector used in the DNA sequencing (Ln 86), pPICZαA (Ln69), 6HP exchange column, CarboPacPAC20, etc.
22. Concerning the construction derivative of the vector pPICZαA obtained in this work (Ln 209) the signal protein peptide was removed and the gene sequence encoding the mature peptide was cloned in pPICZαA fused to the α-factor secretion signal of Saccharomyces verevisiae …to direct the export of the protein to the extracellular yeast medium. To complete the sentences to facilitate the reader's understanding.
33. Unify throughout the entire text whether or not to place commas to indicate units of thousands in the quantities indicated e.g.: 12,273 μmol/min/mg and 7,478.35 16 s-1 (Lns 16-17), 6,300 × g (Ln 158), 4000 U (Ln 148), 1143 bp (Ln207), 1571.67 U/mL (Ln 213, etc.; and by the way, a single decimal is more than enough for amounts greater than 1000.
44. The representation of the Figure used for the calculation of the kinetic constants (Ln 235) should be provided as supplementary material and if any type of program has been used in the calculation, it should be also indicated in section 2.6.
55. Fig 2C is commented in the text before Fig 2B. Renumber the panels of Fig 2 or the reference to the panels indicated in the text.
66. In Figure 2 legend, it is not clear what is considered as 100% activity. To include this information.
77. In analysis of hydrolysis products by HPAEC-PAD and products quantified in Fig 3, show in the supplementary material some representative chromatogram so that we can see if many more products are produced (or not) without being identified due to lack of patterns. Also concerning to Fig 3, no G1 seems to forma and in the line 266 is mentioned: mono-galacturonic acid only accumulated in small amount. Either another Fig is shown in which some formation of G1 is appreciated or this phrase is changed.
88. Lines 276-278, It does not make sense to me since only one type of substrate has been used. Also, I miss some information/discussion on the degree of polymerization of the substrate used in this work, it is only indicated that it is from SIGMA. Is there a lot of material left without hydrolyzing after pePGA treatment? After the reaction carried out by this enzyme, the mixture is ever centrifuged to eliminate the non-hydrolyzed materials..? This was not indicated when use TLC analyses (Ln 149) but it was for the bioactivity tests (section 2.8).
9. Figure 6, in the case of Candida albicans, make some reference in the discussion to the results obtained when ampicillin is used as a positive control.
Reviewer 2 Report
The paper entitled “A novel endo-polygalacturonase from Penicillium rolfsii with prebiotics production potential: cloning, characterization and application” indicates heterologous expression and biological properties of an endo-polygalacturonase. The manuscript has shown reliable results as a work of biological characterization of the recombinant enzyme. Although the manuscript shows significant results, some data in this paper seem to be insufficient, and the findings are not novel. My comments are shown below.
1) Why did the authors use the fungus Penicillium rolfsii in this study? Many GH28 pectinases have already been obtained from Penicillium spp. What is difference between a purpose and results of this study and those of these studies?
2) L109; What is “SDS-buffer”? Please describe in detail.
3) L117, L138; Please provide details (e.g., degree of methylation, manufacturer, raw material) of the polygalacturonic acid used. Do the authors not test the action of this enzyme on other pectin-rich agricultural by-products?
4) The authors should analysis the activity against other substrates (e.g., galacturonic oligo saccharides, sodium polygalacturonate, calcium polygalacturonate gel, esterified pectin, and sodium alginate), to determine the substrate specificity of the endo-polygalacturonase, pePGA.
5) Figure 6; It is not clear why pePGA-POS has antimicrobial activity against bacteria. The authors should also try other enzymatically degraded pectin degradation products, original polygalacturonic acid, galacturonic acid monomer, and other oligosaccharides (such as acidic oligosaccharides).
Round 2
Reviewer 2 Report
I read the revised paper entitled “A novel endo-polygalacturonase from Penicillium rolfsii with prebiotics production potential: cloning, characterization and application” submitted to Foods. I regard that the problems on this manuscript have already been revised.